# Wet Blue Enzymatic Treatment and Its Effect on Leather Properties and Post-Tanning Processes

**DOI:** 10.3390/ma16062301

**Published:** 2023-03-13

**Authors:** Renata Biškauskaitė, Virgilijus Valeika

**Affiliations:** Faculty of Chemical Technology, Kaunas University of Technology, Radvilenu pl. 19, 50254 Kaunas, Lithuania

**Keywords:** wet blue, enzyme preparation, re-bating, leather

## Abstract

Due to their variety, specific activity, and mild reaction conditions, enzymes have a wide application in beam house processes such as soaking, dehairing, bating, and de-greasing. Recently, due to improvements in biotechnology, re-bating after chroming has received increased attention. The aim of this work was to investigate the application of enzyme preparation in the re-bating process and its effect on the semifinished and finished product, as well as its influence on post-tanning operations. The enzymatic treatment of chromed semifinished leather (wet blue) led to a higher shrinkage temperature (1–6 °C), greater water vapour absorption (0.3–5.5%), better chromium compounds exhaustion during re-chroming (4–21%), and better dye penetration. Moreover, collagen was affected during the enzymatic process; the results showed a greater concentration influence in the operation compared to the process time. On the other hand, no effect on the physical and mechanical properties and fat-liquoring process was observed. Overall, these results indicate that some properties and processes are improved; however, before use for re-bating, every enzyme should be carefully investigated.

## 1. Introduction

The leather industry employs waste from the food industry to produce diverse products, such as garments, footwear, upholstery, etc. However, these operations use a significant amount of chemicals and water, producing huge amounts of solid and liquid waste and leading to a negative effect on the environment [1]. To decrease environmental impact, new technologies are being developed; these can be classified into two main groups:(1)Wastewater treatment and cleaner solid waste processing;(2)Cleaner technologies that reduce the pollution load or do not use hazardous chemicals in the processing of leather [2].

Currently, studies are investigating enzyme employment in leather processing, not only to reduce the impact on the environment, but also to improve the properties of the leather. Enzymes find wide applications in beamhouse processes (soaking, dehairing, bating, de-greasing) because of their variety, specific activity, and mild reaction conditions [3].

Despite the improvements in biotechnology, enzymes are mainly used in the bating of hide/skin after unhairing and the opening of derma; for this step, the use of ferments is crucial. During bating, non-collagenous substances such as albumin, elastin, globulin, and proteo-glycan are removed. This step is necessary to create a soft finished product, increase the area, and prepare leather for tanning [4,5,6]. Furthermore, non-collagenous proteins remaining in hide can cause a lack of flexibility and firmness in dried leather [4]. For interfibrillar substances, the removal of alkaline or neutral proteases is carried out; nevertheless, new studies show the possibility of acid protease application in bating [7,8,9,10]. Currently, proteases used in bating are derived from non-hazardous microorganisms.

It has been found that a higher amount of enzyme can remove more non-collagenous substances, which influence physical and chemical properties such as shrinkage temperature, strength, and porosity [11]. During bating, carboxyl groups are activated in the pelts and, because of this, more chromium binds to collagen. According to Nugrahaa et al., when applying crude enzyme, fibre opening depends on the concentration used. A higher dose improves fibre opening in leather; due to this, a higher penetration of tanning substances is achieved by the ability to create bonds with collagen to carboxyl and amine groups [12]. In addition, process time plays an important role; higher protein mass transfer is achieved with a longer reaction time/longer contact between the enzyme and the pelt. However, with process time and concentration, grain damage increases [13].

Further studies have been carried out on the use of acid proteases in bating. The study by Yongquan [9] showed a higher efficacy of acid protease from *Aspergillus usamii* in sheep pelt bating compared to the use of neutral protease. Širvaitytė et al., used acidic enzymes in pelt bating after de-liming with peracetic acid; the process results indicating the suitability of acidic protease in bating [10]. Other researchers have reported promising results in pickling–bating processes [7,14,15], where enzymes were used in a pickling solution after bating, or two processes were combined together.

As a result of improvements in biotechnology, re-bating or wet blue bating has received more attention. The wet blue bating process is an essential step in minimizing the differences of wet blue, which are purchased from different regions; during the process, acid protease is used in the treatment of wet blue to equally distribute collagen fibre and improve the quality of the finished leather [16]. Nevertheless, few studies have been conducted on the use of enzymes during tanning and post tanning, due to their limited activity on leather [17].

The analysis by Li et al., on the re-bating mechanism and effect on wet blue showed different resistance to chromium depending on the origin of the used acidic proteases. The proteases of *Bacillus* had a lower resistance than those of *Aspergillus*. Wet blue after treatment with *Aspergillus* protease had better properties and its collagen fibres were better dispersed. Consequently, elastin degradation was 0.5‰ compared to 0.006‰ collagen; this indicates that the enzyme mostly effected elastin, and the finished product was not damaged [16]. The higher hydrolysis of elastin can be explained by the lower binding capacity of the elastin fibre with Cr. This is because elastin is composed of a low percentage of acidic amino acids [5].

The study by Zhang et al. [15] suggests that bating performance depends on the amount of Cr in the leather. After chroming, the leather becomes more resistant to protease. Cross-linking improves the stability of the semi-product; however, binding with Cr can cause changes in protein molecular conformation. Due to this, enzymes may not be able to identify the action sites, and their catalytic activity will be lower. The results obtained from the study show different activity to the wet blue leather based on different enzymes; process effectiveness correlates with protease activity on chrome-tanned elastin and collagen fibres [18].

Although re-bating can improve the finished product, the process requires a higher enzyme concentration and a longer operating time [7]. As a result, it is important to determine the properties of the leather after re-bating, as well as the effect on the following processes, such as re-tanning, dyeing, and fat-liquoring. The aim of this work was focused on the application of enzyme preparation (EP) in the re-bating process and its effect on the semifinished and finished product, as well as its influence on post-tanning operations.

The novelty of this study’s approach is to establish an effect of treatment by various EPs on changes in the structure of wet blue collagen, and to clarify how and which properties of wet blue and finished leather change dependently (or independently) on the origin of EP used for the treatment and the treatment parameters. Establishing this may provide information for which processes’ EP application can be beneficial.

## 2. Materials and Methods

### 2.1. Materials

Bovine leather that had undergone the chroming process (wet blue leather) was acquired from the Lithuanian tannery “TDL Oda”. The purchased wet blue leather (WB) was processed according to conventional technology, which was valid in the tannery. The leather was split and shaved to a thickness of 1.65–1.70 mm. The main qualitative indexes of WB: amount of Cr_2_O_3_—5.33 ± 0.21%; pH—4.25 ± 0.02; and shrinkage temperature—113.3 ± 0.6 °C. The WB was sliced into a series of samples, so that each experiment would include all parts of the leather. The pieces from the rump part were marked and used for an FTIR analysis, as well as other qualitative indexes such as shrinkage temperature, amount of chromium, strength properties, and water vapour absorption.

The chemicals used for the analysis were of analytical grade, whilst analytical and technical grade materials were used for the technological processes.

For re-bating, four different enzyme preparations (EPs) were used: Zime SB (River Chimica, Italy), isolated from *Aspergillus oryzae*, which is a bating EP for acid bate; NovoBate WB (Novozymes, Denmark), isolated from *Baccilus* microorganisms, which is used in the re-bating process; Oropon DVP and Oropon WB (TFL Ledertechnik GmbH, Rheinfelden, Germany), both EPs, isolated from *Aspergillus Niger*; Oropon DVP, which is used as a bating agent for pickle pelts; and Oropon WB, which is used as a bating agent for wet blue.

Other technical products used for the technological processes were the following: Cromeco 33 Extra (contains 25% of chromium (III) oxide, 33% basicity), produced by Gruppo Chimico Dalton (Limbiate, Italy); Neutragene MG-120, for increasing the chromium compounds’ basicity (Codyeco S.p.a., Santa Croce sull’Arno, Italy); fatliqours Oleal 146, Oleal 1946, Fospholiker 661, and Fospholiker 6146 (Codyeco S.p.a., Santa Croce sull’Arno, Italy); dye Sellaset red H (TFL Ledertechnik GmbH, Rheinfelden, Germany); and mimosa tannins and quebracho tannins (Tanac S.A., Montenegro, Brazil).

### 2.2. Technological Processes

All technological processes were carried out as follows (Table 1). Re-chroming, neutralisation, dyeing, fat-liquoring, and re-tanning were performed according to conventional leather-processing technology.

### 2.3. Analysis Methods

The caseinolytic activity of EP was determined using the Anson method [19]. Sodium caseinate was used as a substrate. For the determination of collagenolytic activity, the modified method of Xian et al. [20] was used. Soaked leather fibres from lyophilized hairless hide were prepared and chosen as a substrate. An amount of 300 ± 1 mg of fibre was accurately weighed in a 20 mL test tube, followed by adding 9 mL of B-R buffer, (5 pH) and stirred in a shaker for 10 min at 40 °C with 200 r/min. Then, 1 mL of enzyme solution (diluted to a certain concentration with the same B-R buffer) was added and stirred for another 30 min at 40 °C with 200 rev/min, precisely. Finally, the reaction liquor was filtered with a qualitative filter paper, and the reaction solution was hydrolysed using 6 N of HCl at 120 °C for 10–12 h. The concentration of hydroxyproline in the digested liquor was tested according to the photo-colorimetric method [21].

The amount of collagen proteins removed was estimated from the amount of hydroxyproline in the pickling solution using a photo-colorimetric method [21]. Samples of the wet blue bating solution after the process were hydrolysed using 6 N of HCl at 120 °C for 10–12 h. A formation of coloured soluble product was based on a reaction of hydroxyproline with p-dimethylaminobenzaldehyde. The absorption was measured with a spectrophotometer GENESYS-8 (Spectronic Instruments, Cheshire, UK) at a 558 nm wavelength.

The shrinkage temperature after the re-bating and post-tanning processes was determined as described in the literature using special equipment and replacing the distilled water with glycerol [22]. An image of the equipment and working principle is presented in Appendix A.

Chromium compound exhaustion was estimated by determining the concentration of chromium in the initial re-chroming solution, and in a mixture of the used re-chroming solution and washing (after re-chroming) solution. The concentration of chromium in solution was determined according to the method described in the literature [22]. The method prescribes oxidation of the chromium presented in the solution into a hexavalent state using hydrogen peroxide, and an analysis of the solution by iodometric titration.

For IR spectroscopy, a Perkin-Elmer FTIR Spectrum GX (Waltham, MA, USA) spectrometer with a horizontal attenuated total reflectance accessory was used. The wavelength interval was 4000–650 cm^−1^; the resolution was 4 cm^−1^, and the scan number was 10 times. Before the IR-spectroscopy analysis, the leather samples were dehydrated with acetone [23].

The exhaustion of the dye was determined by the colorimetric method by measuring the absorbance of the dye solution. The dye absorbance was measured using GENESYS-8 (Spectronic Instruments, Cheshire, UK) at a 495 nm wavelength. Distilled water was used as a solvent. Dye consumptions were calculated using the calibration curve.

The penetration of dye through the hide was evaluated using a special optical microscope with scale (magnification 15 times) MPB-2 (Izyum Instrument Making Plant, Izyum, Ukraine).

The strength properties, water vapour absorption, amount of chrome compounds in the leather, the matter soluble in dichloromethane, and the volatile matter were determined according to the standards [24,25,26,27]. Before the mechanical tests, samples of wet blue were dehydrated (fixed) with acetone [23] and dried in fume board at ambient temperature (23 ± 2 °C) for 24 h. The finished leather samples were placed on a table and dried in a free state for 48 h at a temperature of 23 ± 2 °C. After drying, all the test pieces were stored at a minimum of 24 h prior to testing in the standard climatic conditions at a temperature of 23 ± 2 °C and a relative humidity of 50% ± 5%.

### 2.4. Statistical Analysis

All data were expressed as the average value of measurements performed in triplicate. One sample was used for one measurement. Standard deviations did not exceed 5% for the values obtained.

## 3. Results and Discussion

The conventional bating process uses alkaline enzymes, which are most active at higher pH values. However, due to the relatively low pH value of wet blue, for re-bating, it is important to use enzymes that have proteolytic activity in acidic medium. For this, caseinolytic and collagenolytic activity were assessed for the four EPs that were used in the following bating process (Table 2).

The results show different proteolytic activities depending on the substrate used. All the EPs were more active on casein, whereas their activity on collagen was relatively low. This is not necessarily a negative finding, collagenolytic activity that is too high can cause damage to the leather structure. NovoBate and Zime SB showed the greatest results with both substrates, although the differences between both EPs were significant; Zime SB activity was more than four times lower on casein and almost eight times lower on collagen substrate in contrast to NovoBate.

NovoBate WB had the highest proteolytic activity on both substrates compared to other EPs; its caseinolytic activity was more than 166 times higher compared to the activity of Oropon WB, and the collagenolytic activity was 39 times higher compared to the lowest activity of Oropon DVP. As mentioned before, the lowest activity on casein was obtained using Oropon WB; despite this, the activity on the collagen substrate was closer to the Zime SB value. The lowest collagenolytic activity was shown using Oropon DVP EP.

After the proteolytic activity was evaluated, the EPs were used for the WB treatment. Sixteen re-bating variants were tested for chromed leather bating (Table 3).

It is important to determine the effect on collagen, as greater collagen hydrolysis can have an impact on the physical and mechanical properties of leather [28]. Accordingly, after re-bating, the removed collagen amount and shrinkage temperature were assessed (Table 3). The results show a different effect on collagen depending on the EP that was used in the process. EP Zime SB and NovoBate WB had a greater effect on collagen in wet blue. Increasing these EP concentrations led to a higher amount of removed collagen in the re-bating solution; using 5% EP, the removed collagen amount was more than three times higher compared to using 1%. Process times had less influence than concentration. However, the results indicate that Oropon DVP and Oropon WB had a lower impact on collagen in wet blue; the amount removed was very similar, despite the process time or concentration.

Despite the EP effect on collagen, the shrinkage temperature increased after treatment using Zime SB. The results obtained in Table 3 can be explained in that during re-bating, collagen and other substances in wet blue were affected; they were washed out during this process. Enzymatic bating led to better leather fibre opening. Due to this, the gaps between the fibrils narrowed and the shrinkage temperature increased. After re-bating, all samples had a higher shrinkage temperature than the control sample. An increase in shrinkage temperature was also observed in wet blue after treatment with enzymes used in the work of G.C. Jayakumar et al. [17].

After evaluating the amount of removed collagen and the shrinkage temperature, the re-bating process was repeated, and physical properties such as water vapour absorption, tensile strength, and relative elongation at the strain of 10 N/mm^2^ and at the break were determined. As the EP concentration had a greater influence on the effect on collagen than the process time, the physical properties were assessed for samples that were re-bated for 1 h with EP concentrations of 1 and 5% (Table 4).

The results in Table 4 indicate no apparent dependency on physical properties, such as tensile strength, relative elongation at break, and at the strain of 10 N/mm^2^. However, after enzymatic re-bating, the water vapour absorption increased; this can be explained by fibre opening and the removal of non-collagenous substances during wet blue treatment. Enzymatic re-bating results in an increased area; after wet blue bating, more gaseous water can bind to leather [3].

To evaluate how deeply the enzymatic treatment affected WB, an FTIR analysis was performed. The IR spectra after the EP process with all eight variants and control were recorded and analysed (Figure 1). In the control sample, a broad peak was observed at around 3600–3000 cm^−1^, which is usually associated with adsorbed water (O–H stretching vibrations), as well as hydrogen bonds in amino acids (N–H stretching) [29,30,31]. After enzymatic treatment in the 3600–3000 cm^−1^ range, a higher number of low-intensity peaks were observed; this could be explained by the EP effect on intermolecular water and on hydrogen bonds in amino acids in protein. Furthermore, in all experimental samples, new peaks appeared at around 2989–2981 cm^−1^ and 1078–1055, which can be assigned to –CH_2_/NH_2_ vibrations and C–O/C–O–C stretching, respectively. New peaks indicate an EP effect on WB and the formation of new bonds. Typically, 1100–1000 absorbance bands are assigned to carbohydrate moieties in leather [32,33]; however, carbohydrates did not form during the enzymatic treatment. Instead, EP may have affected the leather by hydrolysing the bonds between collagen and other substances. Nevertheless, the FTIR spectra showed that, regarding the formation of new bonds, the absorbance bands for amide (I), amide (II), and amide (III) remained the same.

After re-bating, other post-tanning operations must be performed to achieve the desired properties of the finished product. In this study, first, re-chroming with all 16 variants was performed, and wet blue properties such as the shrinkage temperature and chromium oxide content in leather were determined. Furthermore, the exhaustion of chromium compounds was estimated (Table 5). Today, chromium is the most used chemical in the tanning process; over 90% of leather goods are produced through chroming. Nevertheless, only 55–70% of the chromium salt is fixed in the leather; the remaining salt is fixed to the effluent [34]. The results in Table 5 reveal the influence of enzymatic bating in re-chroming. Enzymatic bating led to higher chromium exhaustion; using 5% Zime SB consumption of chromium compounds can reach up to 79–80%. The study by Zhang et al., showed a similar effect after a pickling–bating process. Using enzymes for pickling–bating, a higher uptake of chrome and post-tanning chemicals was reached. Fibre relaxation enables other tanning and re-tanning substances to better penetrate the hide [15]. This effect was obtained not only after bating, but also after re-bating [17].

The shrinkage temperature after re-chroming became similar to that of the control samples; the differences that were obtained after re-bating disappeared. This can be explained by the fact that although the consumption of chromium increased, crosslink formation with collagen was similar to that of the control; the remaining chrome oxide may bond to collagen without formatting crosslinks. Only when using Oropon DVP at 1% for 3.5 h was the shrinkage temperature lower than the other variants, even though Cr_2_O_3_ exhaustion was greater than 72%.

To influence the determination of the post-tanning processes, six different enzymatic treatments were chosen and performed (Table 6). These series were chosen because the results of the amount of collagen removed, the shrinkage temperature, and the exhaustion of Cr_2_O_3_ showed a greater concentration influence on re-bating compared to the processing time. Moreover, NovoBate WB was excluded due to too high of an effect on collagen.

Fibre opening during re-bating process should loosen the fibers and due to this improve tanning agents, dye and fat-liquors diffusion into the skin [6]. Few studies have been performed with the aim of evaluating dyeing or fat-liquoring efficiency after using EP in re-bating. Song et al. [35] applied different types of proteases in crust leather dying; study demonstrated better dye absorption in treated leather. Other studies [36,37] with protease in leather dyeing showed high dye exhaustion and better distribution of colour in the finished product.

However, during this study a greater amount of fat-liquoring substances were not obtained in the leather or the exhaustion of dyes during the post-tanning processes (Table 6). The highest dye consumption was achieved with the control sample; despite this, all values are similar. The results indicate that re-bating did not have an influence on dyeing consumption. Fat-liquoring process was analogous to dyeing; in finished product the amount of matter soluble in dichloromethane were similar between experimental samples. No apparent dependence on post-tanning processes was observed despite different effects on removed collagen amount, shrinkage temperature, and chromium exhaustion. Despite similar dye exhaustion, different depth of dye penetration was observed using different EP and its concentrations (Figure 2).

In control sample the total penetration of dyes was lowest. Furthermore, a greater diffusion of dyes was achieved on the flesh side, due to well-known structural differences between the grain and the flesh side; flesh side has more open fiber structure that helps dye penetrate better [38]. A tendency to better diffusion through the flesh side was also observed with Oropon WB, different EP concentrations leaded to a similar effect. Nevertheless, diffusion of dyes on the flesh side with Oropon WB increases significantly compared to control samples. Using 1% Oropon WB achieved 35.4% of dye penetration through flesh and with 5–37%. This indicates that although dye exhaustion was similar, the penetration of substances in the leather was deeper.

Treated with Zime SB and Oropon DVP leather showed different dye penetration tendencies. Using these EP led to better dye diffusion on grain side; wet blue affected by re-bating behaved differently during dyeing than control sample. Using higher concentration of Zime SB led to similar dye penetration in both layers; total diffusion in leather was similar to the control.

Physical and mechanical properties after post-tanning were also determined (Table 7). The control sample grain layer broke with less strain applied compared to other samples; only variant treated with a higher concentration of Oropon DVP grain layer broke at lower value. Furthermore, relative elongation of control sample was also the highest. This can be explained that during re-chroming more chromium oxide bonded to experimental samples, however, bonding might appeared more on the surface and because of that the grain layer is stronger than the strength of the samples. Other mechanical properties did not have apparent dependency based on performed process.

Lastly, FTIR analysis was performed after post-tanning. After enzymatic treatment spectra differences between control sample and experimental were observed; however, after post-tanning processes FTIR spectra are almost identical (Figure 3). This means that differences appeared after re-bating are eliminated during later operations.

## 4. Conclusions

The enzymatic WB treatment led to an additional fibre opening; the process changed the WB properties as well as influenced later processes. During the study, a greater influence of EP concentration was observed compared to the process time. Depending on the enzymes used, the amount of removed collagen varied. Moreover, after the process, all samples showed greater thermostability and higher water vapour absorption compared to the control. Furthermore, the effect of the EP caused changes in the supermolecular structure, which was reflected in the FTIR spectra.

After the enzymatic treatment, WB bonded more chromium in the re-chroming process and deepened the penetration of dyes; using EP Zime SB 1% and Oropon DVP, the dyes penetrated better through the grain layer. On the other hand, there were no significant improvements in the mechanical properties, dye exhaustion, and fat-liquoring process.

Overall, these results indicate that some properties and operations are improved, while other are similar to conventional processing. Nevertheless, every enzyme application in re-bating should be carefully investigated in order to achieve the best process conditions and desired product properties.

## Figures and Tables

**Figure 1 materials-16-02301-f001:**
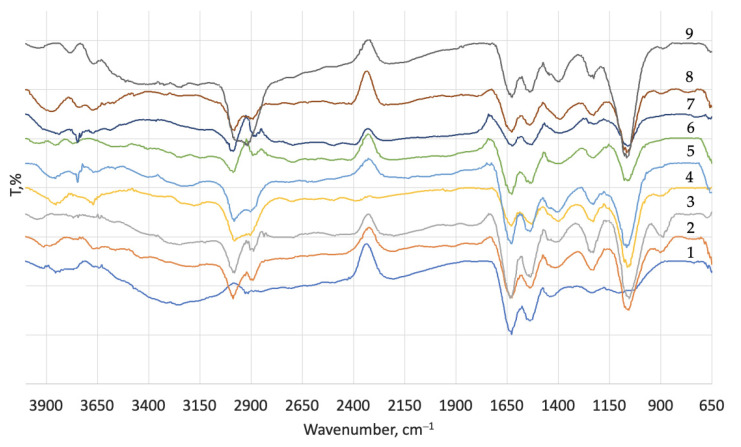
FTIR spectra of untreated wet blue (1) and after enzymatic treatment of wet blue with Zime SB 1% (2); Zime SB 5% (3); NovoBate WB 1% (4); NovoBate WB 5% (5); Oropon DVP 1% (6); Oropon DVP 5% (7); Oropon WB 1% (8); Oropon WB 5% (9).

**Figure 2 materials-16-02301-f002:**
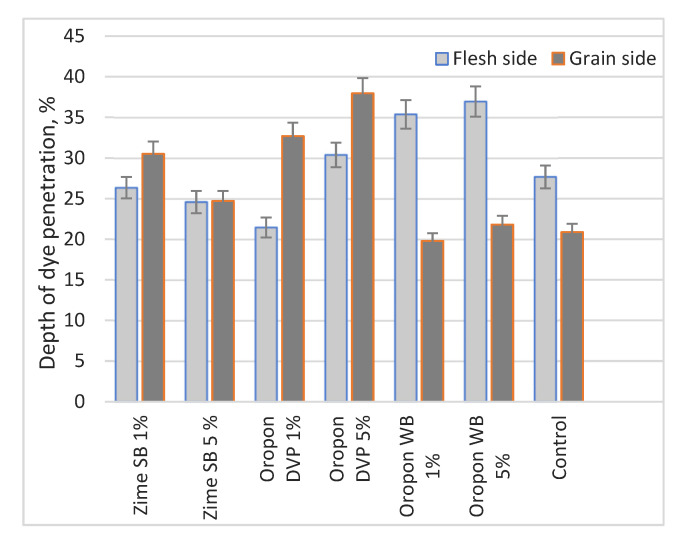
Depth of dye penetration using different EP. The thickness of the wet blue leather samples was within 1.65–1.70 mm.

**Figure 3 materials-16-02301-f003:**
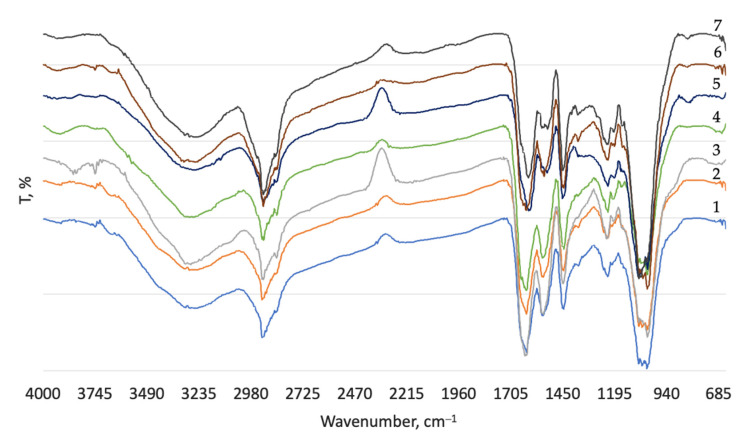
FTIR spectra after post-tanning processes of untreated wet blue (1) and after enzymatic treatment of WB with Zime SB 1% (2); Zime SB 5% (3); Oropon DVP 1% (4); Oropon DVP 5% (5); Oropon WB 1% (6); Oropon WB 5% (7).

**Table 1 materials-16-02301-t001:** Technological processes.

Process	Materials ^1^, %	Process Duration, min	Process Temperature, °C
Washing	H_2_O–300	30	40
Re-bating ^2^	H_2_O–200; EP–1 or 5	60 or 210	
Washing	H_2_O–100	15	
Re-chroming	H_2_O–150 Chromeco 33 Extra–4	30	
	Neutragene MG-120–0.15	10	
	Neutragene MG-120–0.15	50	
Washing	H_2_O–150	30	
Neutralisation	H_2_O–150; NaHCO_3_–1.5	30	
	NaHCOO–2	90	
Washing	H_2_O–100	30	
Washing	H_2_O–200	15	60
Dyeing	Sellaset red H–4.5	60	
Fat-liquoring	Oleal 146–2; Oleal 1946–4; Fospholiker 661–3; Fospholiker 6146–4	90	
	HCOOH–1	30	
Washing	H_2_O–200	15	30
Re-tanning	H_2_O–100; Mimosa tannins–2; Quebracho tannins–2	60	
Washing	H_2_O–100	15	

^1^ % based on WB mass. ^2^ This step was excluded for control samples.

**Table 2 materials-16-02301-t002:** Proteolytic activity at 40 °C; 5.5 pH for caseinolytic and 5 pH for collagenolytic activity.

Enzyme Preparation	Indexes		
	Caseinolytic Activity, U/g	Collagenolytic Activity, U/g	Ratio of Caseinolytic and Collagenolytic Activities
Zime SB	161.73 ± 7.52	3.5 ± 0.16	46.2
NovoBate WB	700.70 ± 27.35	27.3 ± 1.17	25.7
Oropon DVP	8.54 ± 0.41	0.7 ± 0.02	12.2
Oropon WB	4.20 ± 0.17	2.8 ± 0.11	1.5

**Table 3 materials-16-02301-t003:** EP influence on the amount of collagen in the solution and shrinkage temperature.

Enzymatic Treatment				Indexes	
Variant Number	Enzyme Preparation	Amount of EP, % Based on Wet Blue Mass	Duration, Hours	Amount of Removed Collagen, g/kg Wet Blue	Shrinkage Temperature, °C
1	Zime SB	1	1	0.014 ± 0.0004	119.7 ± 0.4
2	NovoBate WB	1	1	0.039 ± 0.001	118.0 ± 0.1
3	Oropon DVP	1	1	0.019 ± 0.0001	119.8 ± 0.3
4	Oropon WB	1	1	0.001 ± 0.0001	115.8 ± 0.3
5	Zime SB	5	1	0.051 ± 0.002	119.4 ± 0.3
6	NovoBate WB	5	1	0.106 ± 0.006	114.0 ± 0.1
7	Oropon DVP	5	1	0.018 ± 0.0001	117.7 ± 0.3
8	Oropon WB	5	1	0.002 ± 0.0001	116.0 ± 0.1
9	Zime SB	1	3.5	0.014 ± 0.0005	119.9 ± 0.3
10	NovoBate WB	1	3.5	0.039 ± 0.0015	116.0 ± 0.1
11	Oropon DVP	1	3.5	0.018 ± 0.0009	118.9 ± 0.3
12	Oropon WB	1	3.5	0.001 ± 0.00004	114.0 ± 0.1
13	Zime SB	5	3.5	0.059 ± 0.002	119.3 ± 0.3
14	NovoBate WB	5	3.5	0.128 ± 0.005	114.7 ± 0.4
15	Oropon DVP	5	3.5	0.020 ± 0.001	117.7 ± 0.4
16	Oropon WB	5	3.5	0.002 ± 0.0001	117.8 ± 0.3
Control	-	-	-	-	113.3 ± 0.6

Note: All variants: water 200%, temperature 40 °C.

**Table 4 materials-16-02301-t004:** Physical and mechanical properties of WB after treatment with EP.

Re-Bating Variant	Relative Elongation of Leather at the Strain 10 N/mm^2^, %	Relative Elongation of Leather at the Break, %	Tensile Strength of Leather, N/mm^2^	Water Vapour Absorption, g/mm^2^
1	25.74 ± 0.78	51.73 ± 1.21	26.10 ± 0.85	20.52 ± 0.69
2	26.14 ± 0.75	51.94 ± 2.07	25.86 ± 0.62	24.20 ± 1.18
3	27.21 ± 0.85	48.51 ± 2.11	23.00 ± 0.60	22.50 ± 0.58
4	26.57 ± 0.43	49.59 ± 0.70	25.32 ± 0.56	25.86 ± 1.21
5	26.15 ± 0.51	52.05 ± 2.11	26.04 ± 0.74	24.80 ± 1.12
6	26.06 ± 0.81	50.39 ± 2.24	25.53 ± 0.82	24.28 ± 0.64
7	26.71 ± 0.64	49.16 ± 2.05	24.93 ± 0.38	22.06 ± 0.37
8	26.53 ± 0.51	48.70 ± 2.34	24.96 ± 0.88	25.48 ± 1.00
Control (without re-bating)	27.08 ± 0.91	51.89 ± 1.01	25.47 ± 0.67	20.32 ± 0.37

**Table 5 materials-16-02301-t005:** Influence of re-bating process on WB re-chroming and shrinkage temperature.

Re-Bating Variant	Indexes		
	Shrinkage Temperature, °C	Exhaustion of Chromium Compounds, %	Cr_2_O_3_ in Leather, %
1	123.8 ± 0.3	68.8 ± 0.4	6.78 ± 0.31
2	122.7 ± 0.4	66.4 ± 0.8	6.80 ± 0.24
3	123.1 ± 0.4	66.5 ± 0.3	6.66 ± 0.23
4	123.5 ± 0.5	67.1 ± 0.6	6.69 ± 0.31
5	124.1 ± 0.4	78.9 ± 0.3	6.78 ± 0.29
6	122.3 ± 0.6	67.6 ± 0.4	6.46 ± 0.13
7	123.0 ± 0.1	62.4 ± 0.8	6.39 ± 0.21
8	121.3 ± 0.3	63.5 ± 0.5	6.59 ± 0.32
9	123.2 ± 0.4	71.6 ± 0.2	6.89 ± 0.24
10	122.2 ± 0.6	70.5 ± 0.7	6.49 ± 0.18
11	118.4 ± 0.3	72.7 ± 0.3	6.54 ± 0.14
12	123.0 ± 0.1	66.9 ± 0.6	6.61 ± 0.21
13	123.1 ± 0.4	80.3 ± 0.7	6.77 ± 0.29
14	123.1 ± 0.4	68.9 ± 0.7	6.46 ± 0.18
15	123.2 ± 0.6	66.9 ± 0.9	6.68 ± 0.27
16	122.0 ± 0.1	62.7 ± 0.8	6.51 ± 0.21
Control (without re-bating)	123.0 ± 0.1	58.4 ± 0.6	6.34 ± 0.24

Note. The amount of Cr_2_O_3_ in WB before re-chroming was 5.33 ± 0.21%.

**Table 6 materials-16-02301-t006:** Dye consumption and amount of matter soluble in dichloromethane (%) in finished leather.

Enzymatic Treatment Parameters			Indexes of Finished Leather	
Variant Number	Enzyme Preparation	Amount of EP, % Based on Wet Blue Mass	Dye Consumption, %	Amount of Matter Soluble in Dichloromethane, %
1	Zime SB	1	86.22 ± 3.73	9.98 ± 0.52
3	Oropon DVP	1	84.92 ± 4.14	9.13 ± 0.45
4	Oropon WB	1	83.42 ± 3.79	9.52 ± 0.47
5	Zime SB	5	82.02 ± 3.92	8.44 ± 0.42
7	Oropon DVP	5	84.92 ± 4.24	9.28 ± 0.46
8	Oropon WB	5	87.82 ± 2.33	8.67 ± 0.35
Control (without re-bating)	-	-	88.19 ± 4.09	9.40 ± 0.46

Note: All variants: water—200%, temperature—40 °C, process duration—1 h.

**Table 7 materials-16-02301-t007:** Physical and mechanical properties after post-tanning processes.

Re-Bating Variant	Relative Elongation of Leather at the Strain 10 N/mm^2^, %	Relative Elongation of Leather at the Break, %	Tensile Strength of Leather, N/mm^2^	Strain When Grain Layer Breaks, N/mm^2^
1	27.09 ± 1.21	61.34 ± 1.84	29.04 ± 1.47	25.92 ± 1.21
3	26.70 ± 1.14	56.53 ± 2.05	31.14 ± 1.43	23.62 ± 1.18
4	29.90 ± 1.30	69.59 ± 3.08	33.73 ± 1.54	24.75 ± 1.20
5	27.07 ± 1.33	64.44 ± 2.64	30.68 ± 1.49	22.08 ± 1.00
7	26.84 ± 1.28	64.18 ± 1.79	33.89 ± 1.63	20.28 ± 1.01
8	27.62 ± 1.36	61.72 ± 1.49	32.68 ± 1.61	25.56 ± 1.27
Control (without re-bating)	31.24 ± 1.56	65.96 ± 2.53	32.14 ± 1.59	21.85 ± 1.07

## Data Availability

Not applicable.

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
