# Peer review of "Wet Blue Enzymatic Treatment and Its Effect on Leather Properties and Post-Tanning Processes"

_materials, 2023, doi:10.3390/ma16062301_

Round 1

Reviewer 1 Report

The topic is an interesting one and I enjoyed learning about it. The authors need to take care in how their methods are defined and results presented however. You will see this through my comments below. One of the most important is that the authors have used formulated products rather than discrete chemicals. This means that the amount of enzyme present will vary between them. They should try to include whatever information about them that is available, and acknowledge where this is not known. Otherwise the paper risks becoming a product comparrison, rather than a scientific investigation into the impacts on the collagen structure.

Intro could have more info

-E.g. proteases are used in acid bating. But what type? Where are they obtained from?Many different versions are found and it would be better if the authors were more specific.

Para for lines 64 - 68. The language needs to be much more cautious about the needs for wet-blue bating. Here is described as a 'crucial process'. I also don't understand how it can 'distribute the collagen fibre more evenly'. I can see that there could be a benefit, but as it is not a common process I don’t think it can be described as crucial. Maybe, would be fairer to say ' bating at wet-blue can help diverse raw materials to behave more uniformly during subsequent processing as these protease enzymes will create space within the protein structure for penetration of chemicals

Line 70 grammar.

Line 76/77 grammar

Methods

Line 94 - 96. More info is required here. What was the pH of the wet-blue? Was it split and shaved to a defined thickness? If so what is it? This is essential information for an academic paper. As the bating is a predominantly surface effect and processing is defined by mass of the hides which are used, thicker hides mean stronger bating action per unit area. Where were the samples taken from?

Line 134 - 136 Given the reference from which the shrinkage temperature measurement is referenced is not in English, some description should be included here.

Line 137 - 140. Same point as directly above. The approach should be specified. E.g. ICP-OES with micros=wave digestions. Was it by titration? If so what were the titres?

Results

Table 2 This would benefit from an additional column highlighting the ratio between casein activity: collagen activity. 

Also, it should be clear that these are products not pure enzymes. Do they have an equivalent amount of enzyme or is it just because the quantities in the formulation? Is there a quoted LVU for these products?

Efforts should be made to identify where the variations come from. The ratio of the two activites seems to be the interesting thing here.

Shrinkage temperature results. Care should be taken in discussing trends here. It's not stated what the control is, but assuming this is an unbated sample. It seems the only conclusion that can be made is that the shrinkage temperature is not significantly adversely affected by the bating. This is all it needs to demonstrate as well so there is no need to seek out any other conclusion. With leather, because all hides are variable, it is possible to have a low std deviation between measurements, but this is not reflective of a real world variation.

Tensile measurements. How was the leather dried? This will have a big effect on elongation and elasticity.

FTIR analysis. Explain why more NH2 vibrations might be expected. Also, wood flour or start are common products used in bating agents. Are these the source of the peaks present? The wood particles will not be able to penetrate substantially and so be picked up on the FTIR as a superficial feature.

Line 262 - 268 Grammar

Line 265. Care needs to be taken when puling out individual results like this. It is not demonstrated by any of the other sampels using this bating agent.

Line 315 Duplicate word

Author Response

Authors of the manuscript “Wet blue enzymatic treatment and its effect on leather properties and post-tanning processes " are very thankful for your questions, remarks and proposals. We would like to present the answers to your questions and inform you about the corrections made in the text of manuscript.

Question. Intro could have more info.

Answer. Thank You for the comment. Of course, there are many sources describing the use of enzymes in leather production, so the Introduction can be significantly expanded. On the other hand, the authors consider that the addition of such information would distract from the main focus of the research, which is the use of enzymes in the treatment of chromed leather. Unfortunately, there is very little information of this kind, so it is complicate to extend the introduction on this issue.

The authors have extended the introduction by adding a sentence on the novelty of the work.  

Q. E.g. proteases are used in acid bating. But what type? Where are they obtained from? Many different versions are found and it would be better if the authors were more specific.

A. The additional data about enzyme preparations were inscribed into the text.

Q. Para for lines 64 - 68. The language needs to be much more cautious about the needs for wet-blue bating. Here is described as a 'crucial process'. I also don't understand how it can 'distribute the collagen fibre more evenly'. I can see that there could be a benefit, but as it is not a common process I don’t think it can be described as crucial. Maybe, would be fairer to say ' bating at wet-blue can help diverse raw materials to behave more uniformly during subsequent processing as these protease enzymes will create space within the protein structure for penetration of chemicals

A. Authors agree that the initial citation is too pretentious. Accordingly, this citation has been replaced by another citation, which we consider more appropriate in this case, from the same source of literature. Authors would be happy to include your sentence, but we have no right to attribute authorship to ourselves, and we do not know the name of the author :).

Q. Line 70 grammar.

A. Grammar was corrected.

Q. Line 76/77 grammar

A. Grammar was corrected.

Methods

Q. Line 94 - 96. More info is required here. What was the pH of the wet-blue? Was it split and shaved to a defined thickness? If so what is it? This is essential information for an academic paper. As the bating is a predominantly surface effect and processing is defined by mass of the hides which are used, thicker hides mean stronger bating action per unit area. Where were the samples taken from?

A. Authors supplemented the information about WB main properties, also included the information about which parts of wet blue were used for experiments in Methods part.

At the same time, authors here are a little confused by the reviewer's remark: ”As the bating is a predominantly surface effect …". Authors may partially agree with the idea when it comes to bating after deliming process, when enzymes remove not only non-collagenous substances, but also remove hair and epidermis breakdown products (scud) from the grain side.  

As writes A. Covington in “Covington A. Tanning Chemistry: The Science of Leather. The Royal Society of Chemistry, UK (2009). P.166” about bating: “Its purpose is to break down specific skin components: usually the non-structural proteins are target”. During non-collagenous substances removal from derma, collagen is also opening up. In WB enzymes effect only those non-collagenous proteins that are left in the derma and, which will be eliminated by the appearance of additional spaces between the collagen fibers. EP effects whole derma and WB thickness may have influence only on the duration of derma penetration. 

Q. Line 134 - 136 Given the reference from which the shrinkage temperature measurement is referenced is not in English, some description should be included here.

A. The image of equipment for shrinkage temperature measurement was prepared and working priciple of the eguipment was described and presented in Supplementary file Figure S1.

Q. Line 137 - 140. Same point as directly above. The approach should be specified. E.g. ICP-OES with micros=wave digestions. Was it by titration? If so what were the titres?

A. Authors inscribed short description of the method from Russian literature source mentioned by Reviewer.

Results

Q. Table 2 This would benefit from an additional column highlighting the ratio between casein activity: collagen activity. 

Also, it should be clear that these are products not pure enzymes. Do they have an equivalent amount of enzyme or is it just because the quantities in the formulation? Is there a quoted LVU for these products?

Efforts should be made to identify where the variations come from. The ratio of the two activites seems to be the interesting thing here.

A. Thank you for your remark. In subchapter "materials" it is described that for rebating different enzyme preparations (EP) and not enzymes were used. Authors in materials included information from which microorganisms' enzymes were isolated. However, we cannot present EP formulation, it is confidential. Unfortunately, not all manufacturers provide the standard activity of enzyme preparations in their descriptions. Moreover, even if the activity value is included, the method of activity determination, which also influences the determined activity value, is not mentioned. For these reasons, the authors themselves determined the caseinolytic and collagenolytic activities of the enzyme preparations.

The ratio of caseinolytic and collagenolytic activities was included into Table 2.

Q. Shrinkage temperature results. Care should be taken in discussing trends here. It's not stated what the control is, but assuming this is an unbated sample. It seems the only conclusion that can be made is that the shrinkage temperature is not significantly adversely affected by the bating. This is all it needs to demonstrate as well so there is no need to seek out any other conclusion. With leather, because all hides are variable, it is possible to have a low std deviation between measurements, but this is not reflective of a real world variation.

A. Authors completely agree with opinion of Reviewer related with shrinkage temperature changes dependently on bating. Also, it was clearly indicated what is the control sample (Table 3).  

Q. Tensile measurements. How was the leather dried? This will have a big effect on elongation and elasticity.

A. Authors added information into the subchapter “Analysis methods” on how the wet blue and finished leather samples were prepared for the physical mechanical tests.

Q. FTIR analysis. Explain why more NH2 vibrations might be expected. Also, wood flour or start are common products used in bating agents. Are these the source of the peaks present? The wood particles will not be able to penetrate substantially and so be picked up on the FTIR as a superficial feature.

A. Before adding FTIR results into the manuscript authors checked enzyme preparation FTIR spectra to make sure that their peaks do not interfere with peaks in leather samples spectra. Also, there are no peaks in spectra of enzyme preparations which can be attributed to peaks in starch IR spectra. The full solubility of enzyme preparations allows conclusion that there are neither wood flour nor caolin. Additionally, the samples for FTIR analysis were thoroughly washed to remove any residues of treatment materials (including remnants of enzyme preparations) before dehydration with acetone and FTIR analysis. Accordingly, the carriers of enzymes in EP should not left in wet blue after washing and have influence on formation of the peaks in the spectra of wet blue.

Q. Line 262 - 268 Grammar

A. The grammar was corrected.

Q. Line 265. Care needs to be taken when puling out individual results like this. It is not demonstrated by any of the other sampels using this bating agent.

A.Thank you for the remark. Research results showed that treatment with all EP showed higher chromium exhaustion compared to the control. Authors wanted to highlight treatment with 5% Zime SB, because using it exhaustion can reach 79-80% (78.9 if treatment duration 1 hour; 80.3-treatment duration 3.5 hours).

Q. Line 315 Duplicate word

A. Thank you, authors corrected it.

Reviewer 2 Report

General comments:

Authors have studied the effect of different enzyme formulations on wet-blue leathers and their influence on rechroming and wet-finishing as well as related parameters. However, they need to include similar such studies as references for enzyme application after tanning process.

Also, they need to address specific comments as follows:

Specific comments:

1.       Authors need to provide General Enzyme names along with Trade names and their standard Activity.

2.       Damage to Collagen is not desirable as it tends to weaken the structure as well as leather. In Abstract also it is mentioned.

3.       Only Marginal improvement in Dye uptake, Chrome uptake etc. obtained with Enzyme treatment, what is the Cost benefit analysis.

4.       Salient outcome as a best improvement obtained with Enzyme treatment could be presented as a Table.

5.       Actual spent liquor needs to be analyzed for activity of Enzymes for Casein, Collagen, Elastic etc. 

Author Response

Authors of the manuscript “Wet blue enzymatic treatment and its effect on leather properties and post-tanning processes " are very thankful for your questions, remarks and proposals. We would like to present the answers to your questions and inform you about the corrections made in the text of manuscript.

General comments:

Question. Authors have studied the effect of different enzyme formulations on wet-blue leathers and their influence on rechroming and wet-finishing as well as related parameters. However, they need to include similar such studies as references for enzyme application after tanning process.

Answer. Of course, there are many sources describing the use of enzymes in leather production, so the Introduction can be significantly expanded. On the other hand, the authors consider that the addition of such information would distract from the main focus of the research, which is the use of enzymes in the treatment of chromed leather. Unfortunately, there is very little information of this kind, so it is complicate to extend the introduction on this issue.

Specific comments:

Q. 1. Authors need to provide General Enzyme names along with Trade names and their standard Activity.

A. Thank you for the remark. Authors included information from which microorganisms' enzymes were isolated. Unfortunately, not all manufacturers provide the standard activity of enzyme preparations in their descriptions. Moreover, even if the activity value is included, the method of activity determination, which also influences the determined activity value, is not mentioned. For these reasons, the authors themselves determined the caseinolytic and collagenolytic activities of the enzyme preparations.

Q 2. Damage to Collagen is not desirable as it tends to weaken the structure as well as leather. In Abstract also it is mentioned.

A. The authors fully share the Reviewer's opinion. It is for this reason that the effect of enzyme preparations on collagen proteins was evaluated, and it was determined which enzyme preparations affect collagen the most.

Q. 3. Only Marginal improvement in Dye uptake, Chrome uptake etc. obtained with Enzyme treatment, what is the Cost benefit analysis.

A. The aim of the work was to investigate effect on semifinished and finish product – cost benefit analysis was not performed.

Q. 4. Salient outcome as a best improvement obtained with Enzyme treatment could be presented as a Table.

A. Authors did not attempt to determine which enzyme preparation and which treatment option is best. The authors sought to determine the effect of the treatment by widest possible range of enzyme preparations on changes (positive or negative) in the properties of the wet blue and finished leather.

Q. 5.Actual spent liquor needs to be analyzed for activity of Enzymes for Casein, Collagen, Elastic etc.

A. The authors question whether they understand this question correctly. When the reviewer writes "actual spent liquor", does he mean the solution of spent liquor after the fatliquoring process?

Reviewer 3 Report

Reviewer comments: Dear Respected Author, many thanks for sharing your Manuscript ID materials-2224620, entitled “Wet blue enzymatic treatment and its effect on leather properties and post-tanning processes".  Regretfully I must reject this paper due to various reasons such as poor presentation of manuscript, and no in-depth discussions on r results. There are so many mistakes at various places which cannot be justified even in revised version as well. However, I advise authors please consider my comments as a positive criticism for your future submission.

1- There are so many papers on the same topic. The novelty of your study is unclear and clearly under question.

2- Table 1. Technological processes. Why you have studied these parameters in this table 1 ? and why not any other ? There is no as such information in the manuscript.

3-  There is no real-time figure about the samples preparation which creates many doubts for readers.  

4- So many tables are presented in results and discussion section. Where are the data in these Tables comes from?

5- Only three figures in the entire manuscript ? Furthermore, these figures are poorly presented for example their legends and labelling are very poor.  The colors schemes in the graphs are not also according to the color blind compatibility.

6- No microscopic images to shows the effect of enzymes on the leather ?

7-To conclude, the information in the current manuscript is provided is too low/minimum and one simply cannot reproduce the presented results.  

8- The current manuscript is more similar to a technical report rather than a research paper.

Thus, taking account the quality of the Materials journal. I have to reject this manuscript.

Author Response

Authors of the manuscript “Wet blue enzymatic treatment and its effect on leather properties and post-tanning processes " are very thankful for your questions, remarks and proposals. We would like to present the answers to your questions and inform you about the corrections made in the text of manuscript.

Question.1-here are so many papers on the same topic. The novelty of your study is unclear and clearly under question.

Answer. Nowadays there are many articles about enzyme application in soaking, dehairing, liming, bating after deliming, however, there are very limited information about enzymes and their preparation usage in the post-tanning processes especially in rebating of chromed leather. Furthermore, there are almost no information about influence of enzymatic process on semi-finished product properties as well as the later influence in wet finishing processes.

Q. 2- Table 1. Technological processes. Why you have studied these parameters in this table 1? and why not any other? There is no as such information in the manuscript.

A. Thank you for the remark. Authors included in technological processes that all post-tanning operations (excepting the rebating) were performed according to conventional leather processing technology.

Q. 3- There is no real-time figure about the samples preparation which creates many doubts for readers. 

A. Authors included information in Methods part about wet-blue, that was used for further processes.

Q. 4- So many tables are presented in results and discussion section. Where are the data in these Tables comes from?

A. All the data comes from the experiments that were described in the “Analysis Methods” part, it is experiments’ results. Also, in the discussion part is indicated what experiments are done and what is determined after that. Tables are used for reader to see results in one place and understand differences between the samples.

Q. 5- Only three figures in the entire manuscript ? Furthermore, these figures are poorly presented for example their legends and labelling are very poor.  The colors schemes in the graphs are not also according to the colour blind compatibility.

A. Authors increased legends’ and axis fonts; in FTIR spectra numbers were assigned to each spectrum. Regarding the number of figures: authors are convinced that the value of the data does not depend on the form in which they are presented, but on the obtained data themselves.

Q. 6- No microscopic images to shows the effect of enzymes on the leather ? 

A. From the previous experiences, authors do not think that microscopic images would be useful.

Q. 7-To conclude, the information in the current manuscript is provided is too low/minimum and one simply cannot reproduce the presented results.

A. We are sorry to hear that, however, it is not clear where information is too minimum. From your previous remark we understood that there was too much information. Could the Reviewer be more specific.

Q. 8- The current manuscript is more similar to a technical report rather than a research paper.

A. Authors respect the Reviewer's opinion on this point but disagree with it completely.

Round 2

Reviewer 3 Report

Accepted.